# A forward Brillouin fibre laser

Gil Bashan[1], H. Hagai Diamandi[1,2], Elad Zehavi[1], Kavita Sharma[1], Yosef London[1,3] & Avi Zadok [1✉]

Fibre lasers based on backward stimulated Brillouin scattering provide narrow linewidths and serve in signal processing and sensing applications. Stimulated Brillouin scattering in fibres takes place in the forward direction as well, with amplification bandwidths that are narrower by two orders of magnitude. However, forward Brillouin lasers have yet to be realized in any fibre platform. In this work, we report a first forward Brillouin fibre laser, using a bare off-the-shelf, panda-type polarisation maintaining fibre. Pump light in one principal axis provides Brillouin amplification for a co-propagating lasing signal of the orthogonal polarisation. Feedback is provided by Bragg gratings at both ends of the fibre cavity. Single-mode, few-modes and multi-mode regimes of operation are observed. The lasing threshold exhibits a unique environmental sensitivity: it is elevated when the fibre is partially immersed in water due to the broadening of forward Brillouin scattering spectra. The results establish a new type of fibre laser, with potential for ultra-high coherence and precision sensing of media outside the cladding.

[1] Faculty of Engineering and Institute for Nano-Technology and Advanced Materials, Bar-Ilan University, Ramat-Gan 5290002, Israel. [2] Currently with the Department of Computer Science and Applied Mathematics, Weizmann Institute of Science, Rehovot 7610001, Israel. [3] Currently with the Applied Physics Division, Soreq NRC, Yavne 81800, Israel. ✉email: Avinoam.Zadok@biu.ac.il

Fibre lasers based on backward stimulated Brillouin scattering (backward SBS) are studied and employed for over thirty years[1,2]. Due to the relatively narrow gain bandwidth of backward SBS in fibres, on the order of 30 MHz, a single longitudinal lasing mode may be selected even in comparatively long fibre cavities of narrow free spectral range. Brillouin fibre lasers are used in microwave-photonic signal generation and processing[3], and in fibre-optic gyroscopes[4]. Backward SBS lasers were also successfully demonstrated in micro-resonator and integrated-photonic platforms, such as cadmium fluoride[5], chalcogenide glass[6], silica[7], silicon[8], and silicon nitride in silica[9].

SBS in optical fibres and waveguides may take place in the forward direction as well[10–13]. Forward SBS couples between two co-propagating optical waves and guided acoustic waves that are predominantly transverse[10–13]. Compared with the backward mechanism, forward SBS typically involves lower-frequency acoustic waves, and its linewidth is typically narrower, reaching hundreds of kHz in bare or polyimide-coated fibres[14]. The narrow gain bandwidths enable the selection of single longitudinal modes of even longer fibre cavities. In addition, the photon lifetime in such long fibre cavities can be even longer than those of phonons, allowing for laser linewidth narrowing with respect to the pump[2].

The realisation of forward SBS lasers in fibres faces a fundamental challenge: Unlike the backward effect, forward SBS within a single spatial guided optical mode leads to phase modulation of a continuous input pump wave and does not provide stimulated amplification of a lasing signal[15]. For that reason, forward SBS lasers are difficult to obtain in standard single-mode fibres. Forward Brillouin lasing can be reached in an inter-modal process, between pump and signal optical fields that co-propagate in distinct spatial modes[16]. Landmark demonstrations of inter-modal forward SBS lasers have been reported in waveguides within suspended silicon membranes[17]. These devices require specialty fabrication capabilities. In addition, the signal photon lifetimes in forward SBS silicon lasers are shorter than those of the phonons, hence they presently do not support optical linewidth narrowing. Forward SBS has been thoroughly investigated in standard[10–13], polarisation maintaining (PM)[18], and specialty nano-structured and photonic crystal fibres[19–21]. However, to the best of our knowledge, forward SBS lasers are yet to be reported in any fibre platform.

In this work we propose and demonstrate a first forward SBS fibre laser. The laser is based on an off-the-shelf, panda-type PM fibre. The lasing fibre is 30 metres long and it is stripped of its protective coating to enhance opto-mechanical interactions. Lasing is driven by inter-modal forward SBS amplification[18]. Continuous pump light is applied along one principal axis of the fibre, and a lasing signal is obtained in the orthogonal state. Feedback for the lasing signal is provided in the form of fibre Bragg gratings (FBGs) at both ends of the fibre. Due to the PM fibre birefringence, feedback is provided to the lasing signal only while the non-resonant pump passes the FBGs with low residual reflections. Several regimes of operation are observed, involving a single guided acoustic mode, few acoustic modes, or with many signals due to multiple modes and inter-mixing among them. The single-mode laser linewidth decreases with the output power and reaches few kHz, limited by thermal drifting of the longitudinal cavity modes in the laboratory environment. The lifetime of signal photons in the fibre cavity is 10 times longer than that of the acoustic waves, hence the forward Brillouin laser holds promise for linewidth narrowing with proper environmental stabilisation. The threshold pump power of the forward SBS laser is on the order of 500 mW. The output power of the laser is currently restricted to 250 μW by the onset of intra-modal backward SBS lasing within the cavity, a limitation that can be mitigated. Extensions of the concept may lead to precision sensors of media outside the fibre cladding. As a first example, we demonstrate the effect of water outside the bare fibre cladding on the laser system. Although the presence of water does not affect the optical cavity, it reduces the forward SBS gain coefficient and raises the lasing threshold. Such sensitivity to the surroundings is unique to the forward SBS mechanism.

## Results

**Principle of operation.** Figure 1a shows the schematic cross-section of a panda-type, PM fibre. Strain rods of silica doped with $B_2O_3$ glass induce permanent birefringence between the slow $\hat{x}$ and the fast $\hat{y}$ principal axes. Figure 1b presents a schematic illustration of the dispersion relation of light guided in the two polarisation modes. We denote the effective indices of the slow and fast states as $n_{s,f}$, respectively, and the difference between them by $\triangle n = n_s - n_f$. The PM fibre also supports a large discrete set of guided acoustic modes that propagate in the axial direction[18]. Each mode $m$ is characterised by a cut-off frequency $\Omega_m$, below which it does not propagate.

The dispersion relation between frequency and axial wavenumber of one guided acoustic mode is illustrated in Fig. 1b. Close to cut-off, the axial wavenumber of the acoustic wave

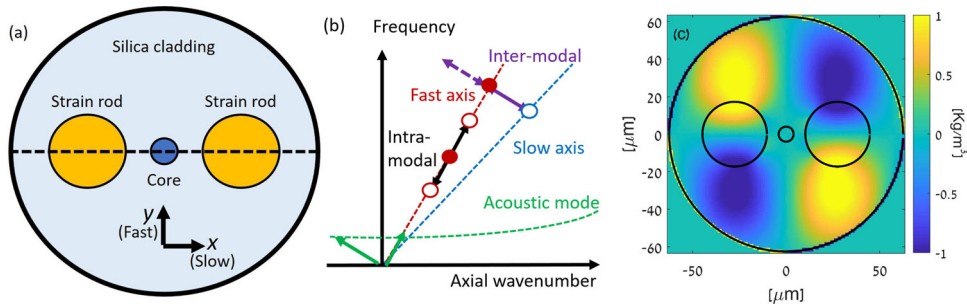

**Fig. 1 Principle of operation. a** Schematic cross-section of a panda-type PM fibre. **b** Illustration of the dispersion relations between temporal frequency and axial wavenumber for light guided in the slow (blue) and fast (brown) principal axes of a PM fibre. The dispersion relation of one guided acoustic mode of the same fibre is shown as well (green). The acoustic mode is characterised by a cut-off frequency, below which it may not propagate. Forward SBS processes can couple light between the two principal axes[18] (purple). Coupling is wavenumber-matched for one sideband only. In this example, light polarised along the fast axis may be downshifted in frequency by the inter-modal forward SBS process. The corresponding upshifting process is hindered by wavenumber mismatch. By contrast, intra-modal forward Brillouin scattering couples an input pump wave to both upper and lower sidebands, leading to phase modulation instead (black). **c** Calculated normalised transverse profile of density oscillations in one guided acoustic mode of a panda-type PM fibre, with a cut-off frequency of 68 MHz.

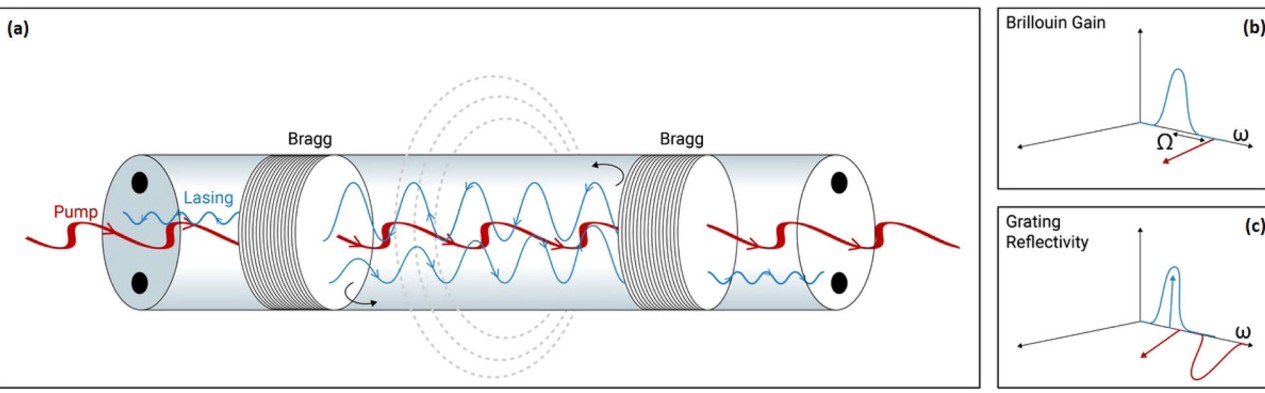

**Fig. 2 Principle of operation. a** Schematic illustration of a forward SBS laser in a PM fibre. A cavity is defined by fibre Bragg gratings at both ends of a bare fibre section. **b** Pump light polarized along the fast axis (brown) provides inter-modal forward Brillouin scattering amplification for light polarized along the slow axis (blue), at a particular optical frequency. The slow axis signal at that frequency is strongly reflected by the gratings. **c** Due the PM fibre birefringence, fast axis-polarised pump light (brown) at a similar optical frequency passes the gratings with low reflectivity.

vanishes, and the mode becomes entirely transverse[10–13]. The material displacement profile of acoustic modes close to cut-off is predominantly transverse as well[10–13]. The axial group velocity of the acoustic wave approaches zero at the cut-off, whereas the axial phase velocity becomes infinitely large. Figure 1c shows the numerically calculated, normalised transverse profile of density oscillations for one guided acoustic mode of a panda-type PM fibre, with a cut-off frequency of 68 MHz.

Consider an optical pump wave of frequency $\omega_p$ that is polarised along the $\hat{y}$ axis. The pump wave can be coupled with a co-propagating, $\hat{x}$ polarised signal wave of frequency $\omega_p - \Omega$ in an inter-modal forward SBS process (Fig. 1b[18]). Coupling takes place through a stimulated acoustic wave of frequency $\Omega$ and axial wavenumber $q_z = \left(-\Delta n \omega_p + n_s \Omega\right)/c \approx -\Delta n \omega_p /c$. Here $c$ is the speed of light in vacuum. The acoustic wave is propagating in the $-\hat{z}$ direction (Fig. 1b[18]). Let us denote the optical power levels of the $\hat{y}$ polarised pump wave and the orthogonal signal wave as $P_{y,x}(z)$, respectively, where $z$ is the axial coordinate. The forward SBS coupling of the two power levels is described by the following coupled equations[18]:

$$\frac{dP_y(z)}{dz} = -2\mathrm{Im}\left\{\sum_m \gamma_m(\Omega)\right\} P_x(z) P_y(z), \qquad (1)$$

$$\frac{dP_x(z)}{dz} = +2\mathrm{Im}\left\{\sum_m \gamma_m(\Omega)\right\} P_x(z) P_y(z). \qquad (2)$$

Here $\gamma_m(\Omega)$ is the coefficient of inter-modal forward SBS through acoustic mode $m$, in units of $[\mathrm{W}^{-1} \times \mathrm{m}^{-1}]$[18]:

$$\gamma_m(\Omega) \approx \gamma_{0m} \frac{1}{1 + 2j(\Omega_m - \Omega)/\Gamma_m}. \qquad (3)$$

In Eq. (3), $\Gamma_m$ is the modal linewidth, which also signifies the decay rate of acoustic energy. The gain coefficient obtains its largest magnitude value $\gamma_{0m}$ at the cut-off, $\Omega = \Omega_m$. $\gamma_{0m}$ is purely imaginary and positive[18], hence Eq. (1) and Eq. (2) describe the Brillouin amplification of the lower frequency, $\hat{x}$ polarised signal wave and the attenuation of the pump[18]. The magnitude $|\gamma_{0m}|$ in uncoated, panda-type PM fibre reaches the order of 1 $[\mathrm{W}^{-1} \times \mathrm{km}^{-1}]$[18]. The strength of the effect on resonance is comparable with that of Kerr nonlinearity.

The inter-modal forward SBS coupling of the pump to an $\hat{x}$ polarised upper sideband signal of frequency $\omega_p + \Omega$ is hindered by a wavenumber mismatch $|\Delta k| = 2n_s \Omega/c$, of about 10 $[\mathrm{rad} \times \mathrm{m}^{-1}]$[18]. This mismatch becomes appreciable over PM fibres that are at least tens of centimetres long. This asymmetry between the coupling of

pump waves to lower and upper sidebands is in marked contrast with forward SBS processes within a single spatial optical mode, such as in standard single-mode fibres[10–13]. Intra-modal forward SBS is characterised by symmetric coupling to both sidebands by the same stimulated acoustic wave, resulting in phase modulation rather than amplitude modulation or amplification (see Fig. 1b[15]). Phase modulation does not reinforce the guided acoustic waves. Therefore, an inter-modal forward SBS process is required to obtain forward SBS gain and lasing.

The proposed forward SBS fibre laser over a PM fibre is illustrated in Fig. 2. A lasing cavity is defined by FBGs at both ends of a bare PM fibre under test. The gratings are designed for peak reflectivity of over 99% for $\hat{x}$ polarised light at frequency $\omega_p - \Omega \approx \omega_p$. Due to the PM fibre birefringence, the same gratings transmit pump light of the same frequency that is polarised along the $\hat{y}$ axis with low reflectivity. The reflection bandwidth of the gratings must be narrow enough to distinguish between the two polarisations. The overall round-trip propagation losses of light polarised along the $\hat{x}$ axis are on the order of 2–3% (see experimental characterisation in the following section.) The inter-modal forward SBS amplification induced by hundreds of milliwatt level pump waves over tens of metres of bare PM fibre can overcome these losses and reach a lasing threshold. The optical frequency of the forward SBS laser would be lower than that of the pump, by a difference $\Omega_m$ that corresponds to a guided acoustic mode for which $|\gamma_{0m}|$ is large.

**Experimental results**. Figure 3a (red and green traces) shows the measured and calculated spectra $2\mathrm{Im}\left\{\sum_m \gamma_m(\Omega)\right\}$ of inter-modal forward SBS in a 30 metres-long, bare PM fibre under test (for the measurement setup and protocols and calculations details see Methods and earlier reference[18]). The spectrum consists of sparse and narrow resonances, with peaks observed at frequencies of 68, 169 and 220 MHz, among others. The magnitude of the forward SBS coefficient at 169 MHz was estimated as $1.4 \pm 0.2 \ \mathrm{W}^{-1} \times \mathrm{km}^{-1}$ (see Methods for the measurement procedure). The linewidths of the spectral peaks are between 200–600 kHz.

Figure 3b shows the reflectivity spectra of the cavity formed by two FBGs, spliced at both ends of the same fibre. The Bragg wavelengths of the two gratings were initially offset by 0.4 nm. Prior to the measurement of Fig. 3b, one of the gratings was strain-tuned in a linear stage with a manual precision micrometre to align the two wavelengths. Such alignment reduces the round-trip propagation losses of potential lasing signals to a minimum. The wavelengths of peak reflectivity for $\hat{x}$ and $\hat{y}$ polarisations

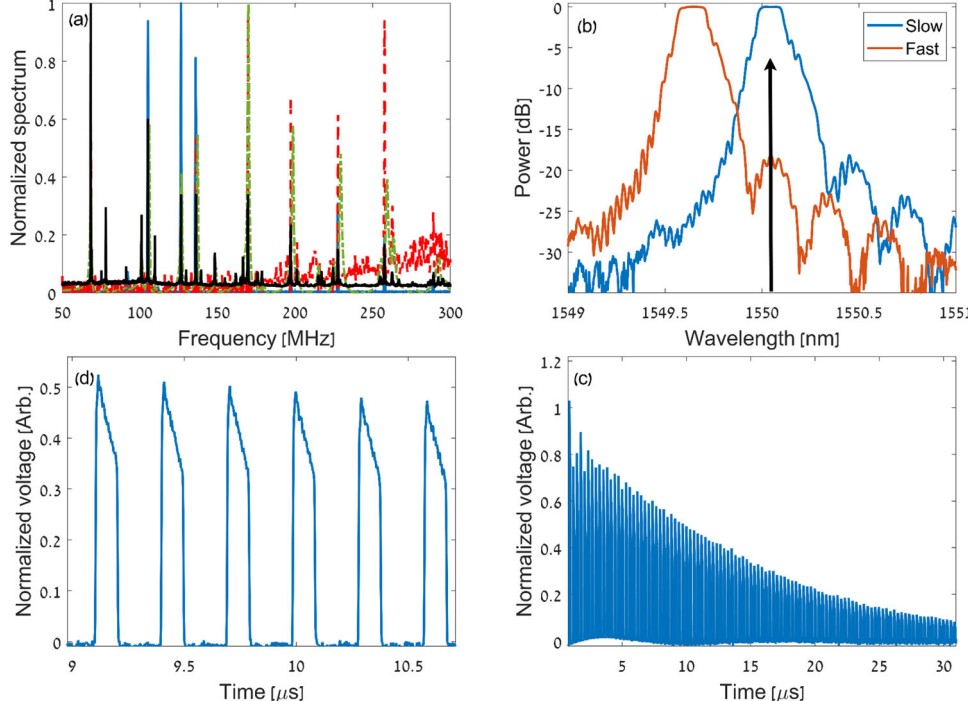

**Fig. 3 Characterisation of forward Brillouin gain and the lasing cavity. a** Measured (dashed red) and calculated (dash-dotted green) normalised spectra of inter-modal forward SBS coupling of power between the two principal axes of a 30 metres-long, bare PM fibre under test. The blue trace shows a beating spectrum between forward SBS lasing and a local oscillator replica of the pump wave, and the black trace shows a corresponding beating spectrum of spontaneous inter-modal forward Brillouin scattering from the laser cavity (see Fig. 5 below and later in the text). **b** Measured reflectivity spectra of the cavity formed by two FBGs, inscribed at both ends of the fibre. Red (blue) traces correspond to light polarised along the fast (slow) principal axes. The black arrow denotes the wavelength of intended forward SBS lasing. At that wavelength, the gratings provide effective reflectivity feedback to light polarised along the slow axis only. **c** Cavity ring down measurement of slow axis polarised light at the wavelength of peak gratings reflectivity. **d** Magnified view of part of the panel **c** trace. The measurements suggest a lifetime of 12 μs, corresponding to round-trip losses of 2.5%.

differ by 0.45 nm (frequency offset of 55 GHz) due to the PM fibre birefringence. The wavelength intended for lasing is noted by the black arrow in Fig. 3b. At that wavelength, the gratings provide effective reflectivity feedback to light polarised along the slow $\hat{x}$ axis only.

Figure 3c presents cavity ring-down measurements of $\hat{x}$ polarised light in the fibre under test. Light from a laser diode of 10 mW power was intensity modulated by repeating pulses of 100 ns duration and 50 μs period in an electro-optic modulator. The laser wavelength was precisely adjusted to match the peak reflectivity of the two FBGs using current and temperature tuning. An isolator at the laser diode output prevented reflections from the cavity gratings from reaching the source. Light at the far end of the fibre was detected by a photo-receiver of 50 V × W⁻¹ responsivity and 2 ns rise time. The output voltage of the detector was sampled by a digitising oscilloscope at 500 Mega samples per second. Traces were averaged over 1024 repeating pulses. The detected trace consists of an infinite series of pulse replicas, within a decaying envelope which represents the lifetime of $\hat{x}$ polarised light in the PM fibre cavity. The lifetime is estimated as 12 μs, corresponding to round-trip losses of 2.5% in the 30 metres-long cavity. The pump power required to obtain the same amount of forward SBS gain is estimated as 600 ± 100 mW.

Figure 4a shows a schematic illustration of the lasing setup. Pump light was drawn from the same laser diode used earlier. The linewidth of the pump laser is specified as 1.25 kHz. The laser output was split in two paths. Light in one arm was amplified in an erbium-doped fibre amplifier (EDFA) of variable output power, aligned with the fast principal axis of the PM fibre under test, and launched into the laser cavity via a polarisation beam splitter. A

bandpass filter and an isolator (not shown) were used to suppress the amplified spontaneous emission of the amplifier and block reflections from reaching back towards the source, respectively. In lasing experiments, the pump wavelength was aligned with that of the gratings peak reflectivity along the slow axis, to provide maximum feedback for potential forward SBS lasing signals. The fast axis polarised pump wave, on the other hand, passes through the gratings with only weak residual reflectivity. In measurements of spontaneous Brillouin scattering, the pump wavelength was detuned from the gratings peak reflectivity by 0.5 nm. With such detuning, light spontaneously scattered to the slow axis received negligible feedback from the FBGs.

The waveform in the second output branch of the laser diode source was aligned with the slow principal axis and served as a local oscillator. The power $P_{LO}$ of the local oscillator was 2 mW. Light emitted from the lasing cavity was mixed with the local oscillator in a PM coupler. Use of PM components outside the fibre cavity enabled beating between lasing signals and the local oscillator without polarisation drifting or fading. The combined output was detected by a photo-receiver of responsivity $\mathscr{R} = 22.5$ V × W⁻¹ and 18 ps rise time. The detector voltage was observed using a real-time digitising oscilloscope and an electrical spectrum analyser. The voltage magnitude $V$ scales with that of the optical field emitted from the laser cavity.

Figure 4b presents an electrical spectrum analyser measurement of the power spectral density $\left|\widetilde{V}(\Omega)\right|^2$ of the detected voltage. The trace was acquired in max-hold mode of operation, over three minutes, with a spectral resolution of 1 MHz. The pump power was 27.5 dBm. A single peak is observed in the beating between the emission from the cavity and the local

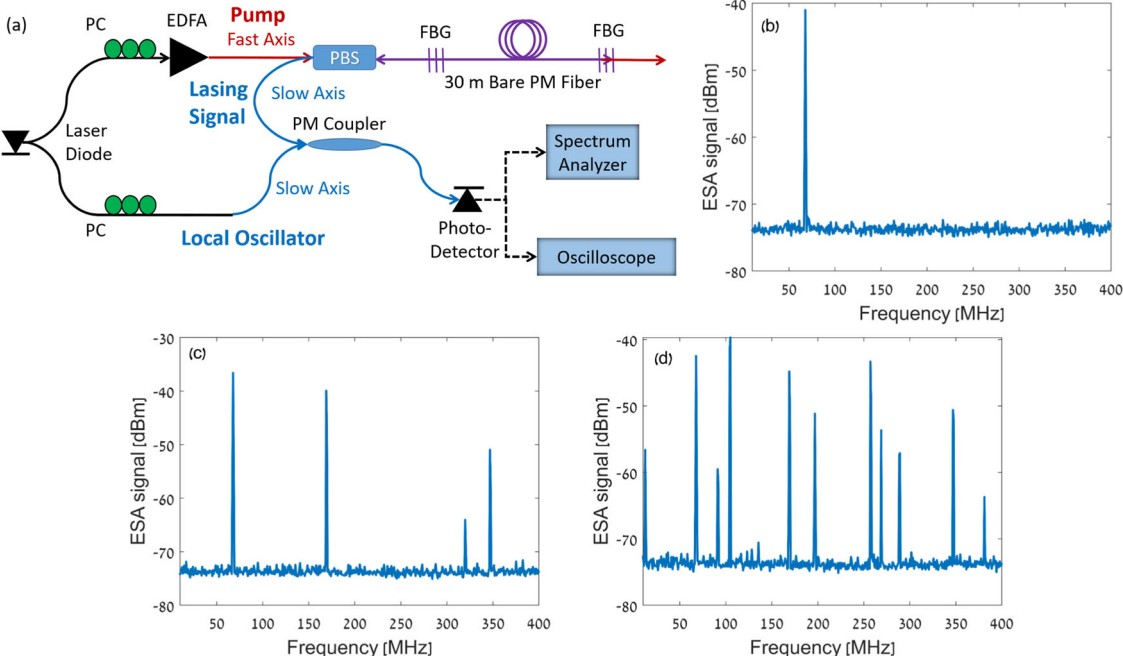

**Fig. 4 Forward Brillouin fibre lasing. a** Schematic illustration of the experimental setup of a forward SBS laser in a PM fibre. EDFA: erbium-doped fibre amplifier; PBS: polarisation beam splitter; PC: polarisation controller; FBG: fibre Bragg grating. **b** Power spectral density (PSD) of the detected beating between the slow-axis output of the PM fibre cavity and a local oscillator replica of the input pump wave. The optical pump power was 27.5 dBm. The trace was acquired by an electrical spectrum analyser (ESA) in max-hold mode over three minutes. A single peak at 68 MHz frequency is observed. This peak corresponds to forward SBS lasing through a guided acoustic mode with that cut-off frequency. **c** Same as panel **b**, with a pump power of 28 dBm. Four peaks are observed, representing forward SBS lasing through acoustic modes of 68, 169, 320 and 347 MHz cut-off frequencies (see Fig. 3a). **d** Same as panels **b** and **c**, with a pump power of 28.5 dBm. A large number of spectral peaks correspond to forward SBS lasing through multiple guided acoustic modes as well as inter-mixing products.

oscillator, at 68 MHz. This frequency matches that of the lowest-order inter-modal forward SBS resonance of the PM fibre under test (Fig. 3a). The results indicate forward Brillouin lasing in the fibre cavity through a guided acoustic mode of 68 MHz cut-off frequency. The acquisition was repeated thousands of times. In most cases, single-frequency forward SBS lasing was obtained through the 68 MHz acoustic mode, however single-peak traces were also acquired at the 169 MHz and 220 MHz frequencies (though with lower probability). Those frequencies also match primary peaks of forward SBS amplification in the fibre under test (Fig. 3a). The forward SBS coefficient at 68 MHz is not the largest of all modes (Fig. 3a). On the other hand, the lower frequency mode benefits from longest acoustic lifetime, which may give preference to lasing at this forward SBS frequency.

When the pump power was raised to 28 dBm, four peaks were observed at 68, 169, 320 and 347 MHz frequencies (Fig. 4c). The three additional frequencies correspond to dominant peaks of inter-modal forward SBS in the fibre (see Fig. 3a). When the pump power was further increased to 28.5 dBm, many spectral components were observed in the lasing signal (Fig. 4d). The peaks are attributed to multiple guided acoustic modes and to nonlinear intermixing among lasing signals.

Figure 5a shows the short-time Fourier transform (STFT) $\left|\widetilde{V}(\Omega, t)\right|^2$ of a 200 ms-long trace of the detected output voltage $V(t)$, where $t$ stands for time. The pump power was 29 dBm. The STFT window was 250 μs long. Forward SBS lasing at multiple frequencies is observed. The lasing peaks switch on and off on a time scale of milliseconds. The dynamics are likely due to the environmental drifting of longitudinal cavity modes through the narrow forward SBS gain peaks. The free spectral range of the longitudinal modes is 3.4 MHz, much wider than the hundreds of kHz linewidths of forward SBS amplification. Temperature

changes on the order of $10^{-4}$ °K are sufficient to scan a longitudinal cavity mode across the forward SBS linewidths[22]. Figure 5b shows part of a $\left|\widetilde{V}(\Omega, t)\right|^2$ trace near 169 MHz, taken again with a pump power of 29 dBm. The instantaneous lasing frequency is drifting over tens of kHz on milliseconds time scales. Drifting restricts the narrowing of the lasing linewidth when observed over milliseconds or longer.

Figure 5c presents the laser output power $P_s$ as a function of the optical power of the input pump wave. Results are shown for forward SBS lasing through the 68 MHz acoustic mode. The laser output power is estimated as $P_s = \left|\widetilde{V}(\Omega)\right|^2 / (2\mathcal{R}P_{LO})$. A clear threshold is identified at a pump power level of 26.5 dBm. The observed threshold is in general agreement with the expected value of $28 \pm 1$ dBm, based on the measured coefficients of inter-modal forward SBS and the cavity round-trip losses (Fig. 3). The output power of the lasing signal saturates at 250 μW, due to the onset of backward SBS in the fibre cavity (see below). The differential slope efficiency of the laser output power between threshold and saturation is approximately 0.005. The low differential efficiency suggests that intra-cavity losses, such as in splicing between fibre sections, are much larger than the losses through the output FBG. Splicing within the cavity may be eliminated with the inscription of FBGs on a single, continuous fibre section.

Figure 5d shows the half-width lasing linewidth, calculated as the standard deviation of $\left|\widetilde{V}(\Omega)\right|^2$, as a function of the output power $P_s$. Results are presented for the same mode of panel (c). The spectra were calculated using STFT with 250 μs-long windows. The linewidths decrease from 6–8 kHz well below threshold towards 2.5–3.5 kHz well above threshold. Further decrease in linewidth was restricted by thermal drifting (see above).

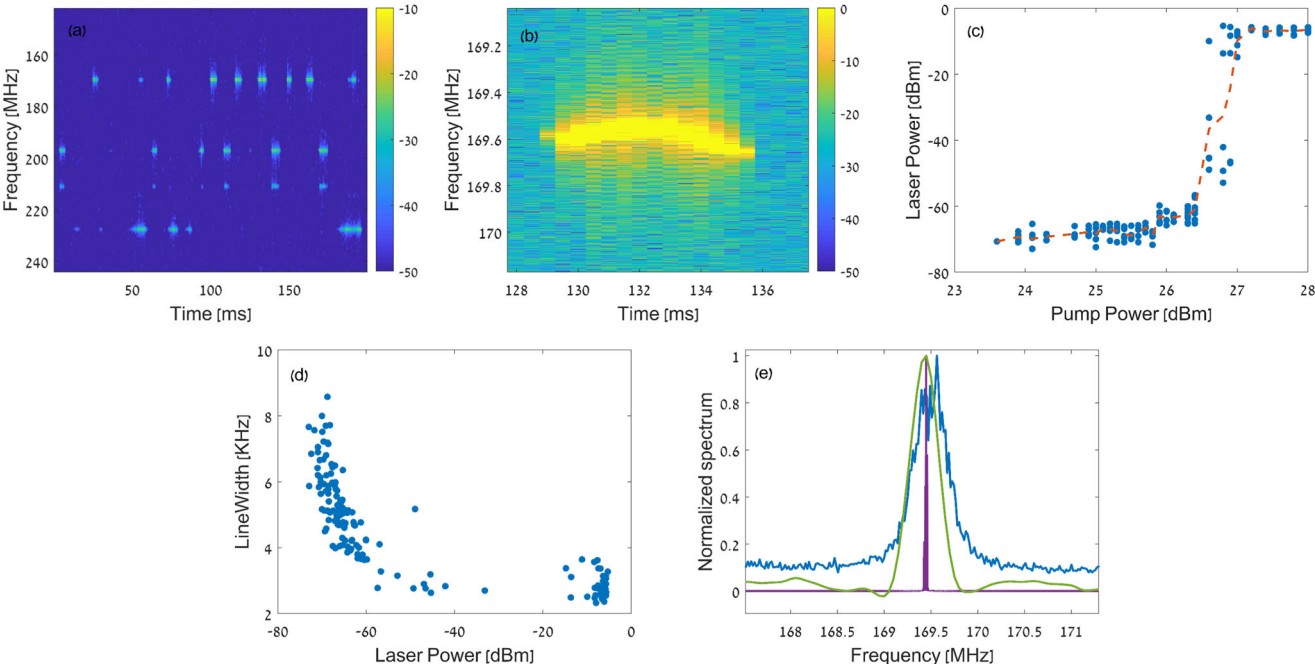

**Fig. 5 Temporal dynamics, threshold, and linewidth of the forward Brillouin fibre laser. a** Relative magnitude (dB scale) of the short-time Fourier transform (STFT) of the detected beating between the lasing signal and the local oscillator over 200 milliseconds. The pump power was 29 dBm. Multiple lasing signals through acoustic modes of different frequencies switch on and off on a time scale of milliseconds, due to environmental drifting of the longitudinal cavity modes across the narrow gain bandwidths of forward SBS. **b** Magnified view of an STFT map similar to that of panel **a**, with the same pump power. The instantaneous lasing frequency through the 169 MHz acoustic mode is drifting by tens of kHz over milliseconds. **c** Measured optical power of the laser output signal as a function of the optical pump power. Results are shown for forward SBS lasing through the 68 MHz acoustic mode. Blue markers denote data points, the red trace represents a trend line. A lasing threshold is observed at a pump power level of 26.5 dBm. **d** Measured half-width linewidth as a function of the laser output power, for the same mode shown in panel **c**. Linewidth were calculated through short-time Fourier transform of detected waveforms, with temporal duration of 250 μs. The linewidths decrease from 6–8 kHz well below threshold to 2.5–3.5 kHz well above threshold. The latter values are transform-limited. Environmental drifting restricts further narrowing of the lasing linewidth when observed over milliseconds or longer. **e** Power spectral density of the beating between spontaneous emission from the fibre cavity through the 169 MHz acoustic mode and the local oscillator (blue, see Fig. 3a). The half-width linewidth of spontaneous inter-modal forward Brillouin scattering is 250 kHz. The green trace shows the spectrum of forward SBS amplification through the same acoustic mode (Fig. 3a). The linewidth of the gain curve is 175 kHz. The magenta trace shows the spectrum of the beating between the lasing signal above threshold and the local oscillator, over a 2 ms-long trace. The pump power was 28 dBm. The beating linewidth is reduced to 5 kHz.

Figure 5e shows the spectra of spontaneous inter-modal forward SBS from the fibre laser cavity (blue trace, see Fig. 3a). The half-width spontaneous scattering linewidth is 250 kHz. The observed linewidth corresponds to an acoustic lifetime of 1.2 μs, 10 times shorter than that of signal photons in the laser cavity. Therefore, the forward SBS fibre laser operates at a regime which may lead to optical linewidth narrowing, with proper stabilisation of environmental drifts. The panel also shows the amplification spectrum of forward SBS through the same acoustic mode (green, linewidth of 175 kHz, see Fig. 3a), and the spectrum of the detected output voltage above threshold (magenta, pump power of 28 dBm). The spectrum was calculated through the Fourier transform of a 2 milliseconds-long trace. The lasing linewidth is 5 kHz, much narrower than those of the amplification spectrum and the spontaneous emission.

Forward SBS lasing in the PM fibre cavity is accompanied by intra-modal backward SBS lasing. The frequencies of backward SBS lasing components are lower than those of the pump and the forward SBS lasing signals by offsets on the order of 11 GHz. Backward SBS lasing along the fast axis is strongly suppressed by the weak cavity reflectivity at that polarisation, below 1% (Fig. 3b). However, backward SBS may reach a lasing threshold along the slow axis. The cavity reflectivity at the backward SBS lasing frequencies along the slow axis is about 40% (Fig. 3b). The cavity feedback is much less effective than that of forward SBS lasing

signals (>99%), however the gain coefficient of backward SBS is significantly stronger, estimated as $200\,\mathrm{W}^{-1} \times \mathrm{km}^{-1}$ in the fibre under test. Consequently, intra-cavity power of tens of mW provides sufficient backward SBS gain to overcome the less efficient gratings feedback and reach lasing in the slow axis through this competing mechanism.

Backward SBS lasing was observed in radio-frequency electrical spectrum analyser measurements of $\left|\tilde{V}(\Omega)\right|^2$ (Fig. 6). Traces were acquired in max-hold mode over 3 min, and the spectral resolution was 1 MHz. A first backward SBS signal is obtained due to partial polarisation crosstalk of the fast axis pump wave to the slow axis. The leaking component benefits from the strong FBGs feedback at the input pump frequency, hence even weak initial polarisation crosstalk builds up within the cavity and reaches the backward SBS lasing threshold. That first backward SBS tone appears at an offset of 10.84 GHz from the pump frequency (Fig. 6a), in agreement with the Brillouin frequency shift in standard fibres[23].

Moreover, the forward SBS lasing signals themselves may reach sufficient power levels within the cavity to become effective pump waves for additional backward SBS lasing components. Figure 6b shows two additional backward SBS signals of 10.91 and 11.01 GHz offsets, driven by forward SBS lasing tones through the 68 and 169 MHz acoustic modes. The gain bandwidth of backward SBS is 30 MHz, much wider than the 3.4 MHz free spectral range of the

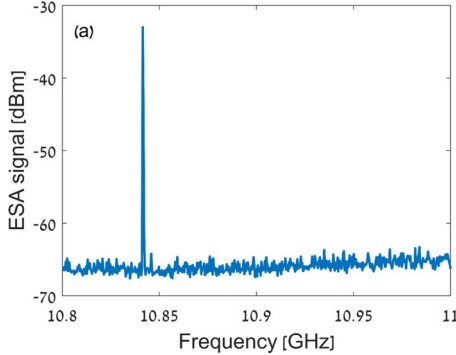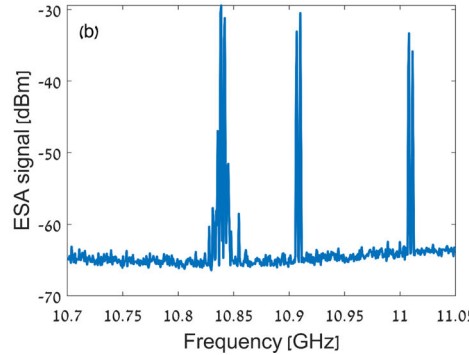

**Fig. 6 Accompanying backward Brillouin lasing. a** Power spectral density of beating between the laser cavity output and the local oscillator, measured by an electrical spectrum analyser (ESA). The ESA acquired data in max-hold mode for 3 min. The pump power was 26.3 dBm. A signal at 10.84 GHz frequency is due to backward SBS lasing in the PM fibre cavity, driven by residual leakage of the input pump wave from the fast principal axis to the slow axis. **b** Same as panel **a**, for a pump power level of 28.4 dBm. Additional backward SBS lasing peaks are observed at 10.91 GHz and 11.01 GHz offset frequencies. These signals are driven by forward SBS lasing components through the acoustic modes of 68 MHz and 168 MHz frequencies.

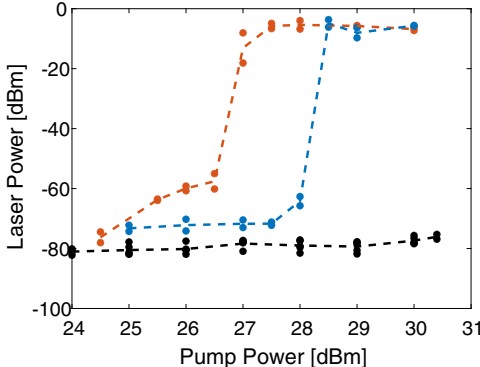

**Fig. 7 Water outside the cavity.** Measured optical power of the laser output signal as a function of the optical pump power. Results are shown for forward SBS lasing through the 68 MHz acoustic mode. The bare fibre cavity was kept in air (red), immersed in water for 40% of its length (blue), or fully immersed in water (black). Markers show data points and dashed traces represent trend lines. Partial immersion elevated the lasing threshold by 1.5 dB, since the forward SBS coefficient in the immersed section is reduced by an order of magnitude. Lasing could not be reached with the fibre was wet for its entire length.

longitudinal modes in the 30 metres long fibre cavity. Lasing at few longitudinal modes is therefore observed within each backward SBS signal. For sufficiently high pump powers, large number of both forward and backward SBS components and inter-mixing among them was observed. The onset of backward SBS lasing limits the intra-cavity power of the forward SBS laser components to tens of mW, and their output powers to only 250 μW. Possible means to modify the balance between forward and backward SBS lasing are discussed in the next section.

The forward SBS fibre laser system is sensitive to media outside the cladding boundary, even though such media does not affect the optical cavity. To illustrate this unique property, the bare fibre cavity was partially or fully immersed in water. Water outside the cladding broadens the spectral peaks of forward SBS and reduces the maximal gain coefficients by an order of magnitude[24]. The immersed sections therefore contribute little to forward SBS amplification along the cavity. As shown in Fig. 7, the lasing threshold was elevated by 1.5 dB when 40% of the fibre length were immersed in water. Lasing could not be reached when the fibre was wet for its entire length. Such sensitivity to conditions outside the cavity is unique to the forward SBS mechanism. The results illustrate the potential of the forward SBS fibre laser in sensing applications.

## Discussion

A forward Brillouin laser was demonstrated using an inter-modal process in an off-the-shelf, panda-type PM fibre. The results establish a first forward Brillouin fibre laser. The lasing cavity was defined by fibre Bragg reflectors at both ends of a 30 metres-long fibre. The threshold pump power was about 500 mW, in agreement with expectations. The output power was 250 μW. Regimes of a single acoustic mode, few modes, and multiple modes were observed. Lasing relied on the asymmetric characteristics of inter-modal forward SBS, in which one sideband of an input pump wave is effectively amplified whereas the generation of the other is hindered by wavenumbers mismatch[16–21]. This property is in contrast with intra-modal forward SBS processes, such as in standard single-mode fibres, where the same stimulated acoustic wave induces two sidebands without symmetry breaking and leads to phase modulation behaviour rather than gain[15].

The forward SBS laser linewidths are on the order of 2.5–3.5 kHz, when observed over 250 μs. These linewidths are transform-limited, and they are 50 times narrower than the modal gain bandwidths of the forward SBS process[14,18], as well as the spontaneous emission bandwidth. When observed over longer integration times, the lasing linewidths are broadened by environmental drifts of the longitudinal modes of the long fibre cavity. The drifts also switch the lasing signals on and off, as the modes pass through the gain bandwidths which are narrower than their free spectral range. Better stability can be obtained using an even longer cavity, in which the free spectral range would match the gain bandwidth, as in many backward SBS fibre lasers[25].

The lifetime of photons in the cavity was measured as 12 μs using cavity ring-down. The lifetime of phonons was observed to be shorter: on the order of 1 μs or less. With this hierarchy between lifetimes, we may expect a narrowing of the laser linewidth with respect to that of the pump wave[2,17,26]. If the cavity length is extended, the photons lifetimes would increase further whereas that of the phonons would remain unchanged. Linewidth narrowing could not be reached in the current experiment due to environmental drifting (see above). Note also that demonstrated forward Brillouin lasers in suspended silicon membrane waveguides did not operate in this regime: the lifetimes of phonons in those lasers are longer than those of the lasing photons[17].

The modest output power reported in this work is compromised by accompanying backward SBS lasing. A first backward SBS tone is driven by polarisation crosstalk of the input pump wave. This contribution may be avoided using better PM components with weaker leakage. Additional backward SBS tones are stimulated by the forward SBS lasing signals themselves.

The onset of these tones may be elevated to higher intra-cavity powers with a more favourable balance between the forward and backward effects. FBGs with sharper reflection spectra would further suppress the feedback provided to backward SBS components, and fibres with a smaller cladding would enhance forward SBS. Lastly, feedback for backward SBS may be avoided altogether in a fibre ring cavity configuration, with an embedded isolator. A longer PM fibre would be necessary in that case, to compensate for dB-scale insertion losses of various components. Intra-cavity power would be limited eventually to several Watts, by amplified spontaneous backward SBS over a single pass, even in the absence of gratings feedback. That threshold may be pushed higher with acoustic anti-guiding at the core of the fibre[27]. Unlike silicon waveguide based lasers[17], the intra-cavity power level would not be restricted by two-photon absorption.

Losses inside the fibre cavity were dominated by those of splices between FBGs and the main, 30 metres-long fibre section. These losses may be eliminated in future realisations through the inscription of gratings on a single, continuous fibre. The pump power at the lasing threshold would be reduced accordingly. The threshold may be reduced significantly in ring cavity configurations in which both pump and signal can be resonant. While a bare fibre was used in this work, similar forward SBS amplification can be obtained in commercially available PM fibres coated with thin layers of polyimide[14].

Forward Brillouin scattering is affected by the mechanical properties of substances outside the cladding[24]. This property serves as the basis of forward Brillouin fibre sensors[14,24,28–31]. Forward SBS fibre lasers can be extremely sensitive to changes in media outside the cladding and/or coating, or to the characteristics of the coating itself. As a first demonstration of environmental sensitivity, we showed that the lasing threshold was elevated when the fibre cavity was partially immersed in water. Similar to corresponding backward SBS configurations[32], forward SBS laser-based sensors may become orders of magnitude more sensitive than feedback-free setups.

In conclusion, a new type of fibre laser has been proposed and demonstrated for the first time, based on inter-modal forward SBS in an off-the-shelf panda-type PM fibre. The principle can lead to highly coherent laser sources and ultra-precision forward SBS sensors.

## Methods
The methods below were already described in our earlier works[18]. They are repeated here briefly for completeness and the convenience of the reader.

### Measurement of inter-modal forward stimulated Brillouin scattering spectra in a polarisation maintaining fibre.
The setup for forward Brillouin characterisation of PM fibres is shown in in Fig. 8. Two Brillouin pump waves were drawn from a common laser diode source at the 1550 nm wavelength range. One pump wave was upshifted in frequency by $\Omega_{IF} = 2\pi \times 9$ GHz using a single-sideband modulator. The intensity of the first pump wave was modulated at a low frequency $f_1 = 2\pi \times 50$ kHz. The pump was amplified by an EDFA to 500 mW power and applied to the fibre under test along the fast axis through a polarisation beam splitter. The second pump wave was spectrally offset by a variable shift $\Omega_{IF} + \Omega$ using a second single-sideband modulator, intensity modulated at frequency $f_2 = 2\pi \times 40$ kHz, amplified to 500 mW power, and launched along the slow axis of the fibre under test. At the far end of the fibre, the slow axis pump wave was detected by a photo-receiver. Forward Brillouin coupling between the pump waves was quantified through monitoring the detected signal at the difference frequency $f_1 - f_2$[33].

### Experimental estimate of the nonlinear coefficient of inter-modal forward SBS.
The input power levels of optical waves taking part in inter-modal forward SBS are denoted as $P_{x,y}(t) = \bar{P}_{x,y}\left[1 + \beta_{1,2}\cos\left(f_{1,2}t\right)\right]$ (see also previous subsection). Here $\beta_{1,2}$ represent the known modulation depths of the two fields and $t$ denotes time. The output power of the slow axis field is modulated at the

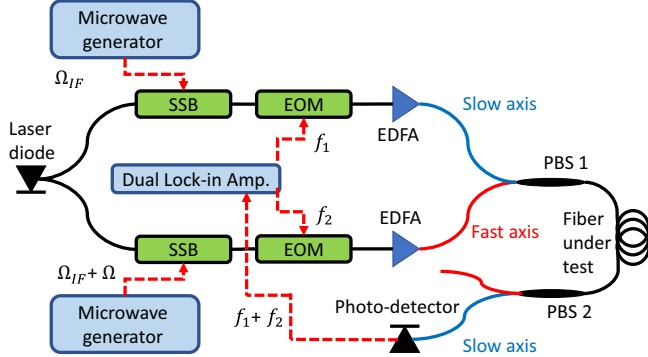

**Fig. 8 Experimental setup for measurements of inter-modal forward Brillouin Scattering[18].** PBS: polarisation bean splitter; EOM: electro-optic amplitude modulator; SSB: single-sideband modulator; EDFA: erbium-doped fibre amplifier.

difference frequency $f_1 - f_2$ due to forward SBS, with a modulation depth of:

$$\beta_{1,out} = \frac{1}{2}\beta_1\beta_2\bar{P}_y\left[2\text{Im}\left\{\sum_m \gamma_m(\Omega)\right\}\right]L. \qquad (4)$$

Here $L$ is the length of the PM fibre. The modulation at $f_1 - f_2$ frequency is detected by the lock-in amplifier, and its amplitude located $V_{out}$ is recorded.

The nonlinear coefficient of inter-modal forwards SBS is estimated using the following protocol. An input wave of mean power $\bar{P}_x$ is modulated at frequency $f_1 - f_2$ using a signal of controllable magnitude $V$ and a modulator with pre-calibrated $V_\pi$. The modulated optical wave is detected and monitored. The voltage $V$ is varied until the detector reading matches the earlier value $V_{out}$ of the forward SBS measurement. We denote that modulating voltage as $V_{ref}$. As the modulation depth matches that of the forward SBS experiment, we may find[18]:

$$J_1\left(\pi\frac{V_{ref}}{V_\pi}\right) \approx \pi\frac{V_{ref}}{V_\pi} = \beta_{1,out} = \frac{1}{2}\beta_1\beta_2\bar{P}_y\left[2\text{Im}\left\{\sum_m \gamma_m(\Omega)\right\}\right]L. \qquad (5)$$

Here $J_1$ is the first-order Bessel function of the first kind, and $V_{ref} \ll V_\pi$. Equation (5) leads to:

$$\text{Im}\left\{\sum_m \gamma_m(\Omega)\right\} = \frac{\pi V_{ref}}{V_\pi \beta_1 \beta_2 \bar{P}_y L}. \qquad (6)$$

### Numerical analysis of inter-modal forward stimulated Brillouin scattering in polarisation maintaining fibres.
Guided acoustic modes of PM fibres were solved through numerical analysis of the elastic waves equation[34]:

$$\Omega^2\vec{u}_m(x,y) + v_S^2(x,y)\nabla^2\vec{u}_m(x,y) + \left[v_L^2(x,y) - v_S^2(x,y)\right]\nabla\left[\nabla\bullet\vec{u}_m(x,y)\right] = 0. \qquad (7)$$

Here $x,y$ are transverse coordinates, $v_{L,S}(x,y)$ represent acoustic velocities of dilatational and shear waves, and $\vec{u}_m(x,y)$ stands for the normalised transverse profile of material displacement in acoustic mode $m$. The parameters of silica were[35]: $v_L = 5,996$ m × s$^{-1}$, $v_S = 3,740$ m × s$^{-1}$, outer diameter of 125.5 μm, and density $\rho = 2,200$ kg × m$^{-3}$. The parameters of the B$_2$O$_3$-doped silica strain rods were[36]: $v_L = 4,895$ m × s$^{-1}$, $v_S = 4,100$ m × s$^{-1}$, and $\rho = 2,080$ kg × m$^{-3}$. The radii of the rods were 17.25 μm, and their centres were located ±27.5 μm away from the fibre axis. The optical mode field diameter was 10.4 μm.

Acoustic decay rates in different media were approximated as: $\Gamma(\Omega) = \Gamma_0 + \Gamma_2\Omega^2$[37]. The coefficients $\Gamma_{0,2}$ were fitted from experiment. For silica we found: $\Gamma_0 = 1.25 \times 10^6$ rad × Hz, $\Gamma_2 = 1.6 \times 10^{-12}$ rad$^{-1}$ × Hz$^{-1}$, and for the strain rods: $\Gamma_0 = 7.25 \times 10^6$ rad × Hz and $\Gamma_2 = 1.6 \times 10^{-12}$ rad$^{-1}$ × Hz$^{-1}$. Modal linewidths were estimated based on the relative confinement of acoustic energy:

$$\Gamma_m = \iint \Gamma(\Omega_m, x, y)\rho(x,y)\left|\vec{u}_m(x,y)\right|^2 dxdy \bigg/ \iint \rho(x,y)\left|\vec{u}_m(x,y)\right|^2 dxdy. \qquad (8)$$

Here $\rho(x,y)$ is the local density.

## Data availability
The source data underlying Fig. 1c, 3, 4b–d, 5, 6 and 7, are provided as a Source Data file. https://figshare.com/articles/dataset/Data_Fig_A_Forward_Brillouin_Fibre_Laser_csv/19927028.

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

## Acknowledgements

This research was supported in part by the Israel Ministry of Science and Technology, Grant no. 61047. Gil Bashan is supported by the Adams Fellowship Program of the Israel Academy of Sciences and Humanities. Hilel Hagai Diamandi is grateful to the Azrieli Foundation for the award of an Azrieli Fellowship.

## Author contributions

G.B. proposed the idea and initiated the project. G.B., H.H.D., Y.L. and A.Z. performed mathematical analysis. G.B., H.H.D. and Y.L. carried out numerical calculations. G.B., H.H.D., K.S., Y.L. and A.Z. designed the experimental setup. G.B., H.H.D., E.Z. and K.S. collected experimental data. G.B., H.H.D. and E.Z. analysed experimental data. A.Z. wrote the draft of the manuscript. All authors commented on the manuscript draft. A.Z. managed the project.

## Competing interests

The authors declare no competing interests.
