## [Peer review file · Nature Communications]

REVIEWERS' COMMENTS

Reviewer #1 (Remarks to the Author):

I have read and carefully analyzed the authors' responses to all reviews from the referees and the authors have responded well to all points. The revised manuscript addresses most of my initial concerns and is now supported by more data and discussion. The authors have now covered all of the items on the laser checklist. I must say however that I am not completely convinced by the performances of this forward Brillouin laser in terms of output power and threshold. Nevertheless, I think the authors have done a very good job responding to most of my comments and I recommend that this revised paper be published in Nature Communications.

Response to Reviewer #1:

"I have read and carefully analyzed the authors' responses to all reviews from the referees and the authors have responded well to all points. The revised manuscript addresses most of my initial concerns and is now supported by more data and discussion. The authors have now covered all of the items on the laser checklist. I must say however that I am not completely convinced by the performances of this forward Brillouin laser in terms of output power and threshold. Nevertheless, I think the authors have done a very good job responding to most of my comments and I recommend that this revised paper be published in Nature Communications."

Reply: We thank the reviewer for his/her positive assessment of the revised manuscript. The present limitations on the threshold and output power are not fundamental, and several avenues for improvement are described in the Discussion section.